# Isotherm, Thermodynamics, and Kinetics of Methyl Orange Adsorption onto Magnetic Resin of Chitosan Microspheres

**DOI:** 10.3390/ijms232213839

**Published:** 2022-11-10

**Authors:** Lina Yu, Jie Bi, Yu Song, Mingqing Wang

**Affiliations:** Food Engineering Innovation Team, Shandong Peanut Research Institute, Qingdao 266100, China

**Keywords:** magnetic resin of chitosan microspheres, methyl orange, physical properties, swelling rates, stability of magnetic particles, batch adsorption, adsorption isotherm model, thermodynamics, kinetics

## Abstract

Severe environmental pollution problems arising from toxic dyestuffs (e.g., methyl orange) are receiving increasing attention. Therefore, dyes’ safe removal has become a research hotspot. Among the many physical–chemical removal techniques, adsorption using renewable biological resources has proved to be more advantageous over others due to its effectiveness and economy. Chitosan is a natural, renewable biopolymer obtained by deactivated chitin. Thus, the magnetic resin of chitosan microspheres (MRCM), prepared by reversed-phase suspension cross-linking polymerization, was used to remove methyl orange from a solution in a batch adsorption system. The main results are as follows: (1) The results of physical and swelling properties of MRCM indicated that MRCM was a type of black spherical, porous, water-absorbing, and weak alkali exchange resin, and it had the ability to adsorb methyl orange when it was applied in solutions above pH 2.0. (2) In batch adsorption studies, the maximum adsorption capacity was obtained at pH 5; the adsorption equilibrium time was 140 min; and the maximum adsorption was reached at 450 mg/L initial concentration. (3) Among the three isotherm adsorption models, Langmuir achieved the best fit for the adsorption of methyl orange onto MRCM. (4) The adsorption thermodynamics indicated that the adsorption was spontaneous, with increasing enthalpy, and was driven by the entropy. (5) The pseudo-second-order kinetics equation was most suitable to describe the adsorption kinetics, and the adsorption kinetics was also controlled by the liquid–film diffusion dynamics. Consequently, MRCM with relatively higher methyl orange adsorption exhibited the great efficiency for methyl orange removal as an environment-friendly sorbent. Thus, the findings are useful for methyl orange pollution control in real-life wastewater treatment applications.

## 1. Introduction

Severe environmental pollution problems arising from toxic dyestuffs are receiving increasing attention [1,2,3]. These non-biodegradable and highly toxic dyestuffs are harmful to humans, even at low concentrations [4,5]. The direct emission of wastewater from dyestuff manufacturers and dye workshops into public drainage systems is prohibited if the wastewater has no appropriate pretreatment. For instance, the water system’s color material might significantly affect the penetration of light and photosynthesis by some aquatic organisms [6,7]. Dyestuffs, a main component of laboratory wastewater, must be purified before environmental discharge.

Methyl orange is a water-soluble azo dye widely used in textiles, printing, paper manufacturing, pharmaceuticals, the food industry, and laboratories [8,9,10,11,12]. In analytical chemistry, it is used mainly as an acid-base indicator because of its sensitivity in aqueous solution with pH of 6.5 (5 g/L, 20 °C) [13]. Methyl orange azo dye is a carcinogenic organic chemical. Like many other dyes, if ingested, methyl orange metabolizes into aromatic amines by intestinal microorganisms. In addition, reductive enzymes in the liver can also catalyze the reductive cleavage of the azo linkage to produce aromatic amines, leading to intestinal cancer [14]. Although methyl orange toxicity has not been quantified, its high content in living systems could be harmful. Therefore, the current research focuses on its safe removal via adsorption.

Over the last decade, physical–chemical techniques, such as electrochemical treatment, photochemical method, adsorption, and biodegradation, were established to remove dyes from wastewaters [15,16,17,18]. Among these, adsorption has proved to be more advantageous over others due to its effectiveness and economy. Investigations have shown that removing toxic and harmful substances using renewable biological resources is a developing approach for environmental protection [19,20].

Chitosan is a natural, renewable biopolymer obtained by deactivated chitin. Chitosan is composed mainly of β-(1-4)-2-amino-2-deoxy-D-glucopyranose (>60%) and β-(1-4)-2-acetamido-2- deoxy–D-glucopyranose repeating units. In recent years, magnetic resins of chitosan microspheres (MRCM) have attracted increasing attention, owing to their unique magnetic features, good biocompatibility, and low toxicity. For example, some scholars have studied the magnetic microspheres prepared by alginate and chitosan, which have superior magnetorheological characteristics [21]. In addition, a study reported the application of magnetic chitosan microspheres for immobilization of Candida rugosa lipase [22]. The thermal stability, storage stability, and conversion of the immobilized lipase were better than those of the free lipase, indicating that the magnetic chitosan microspheres had better biocompatibility and non-toxic properties. In view of this, magnetic chitosan microspheres are extremely valuable in medicine (e.g., drug delivery) and separation (adsorption) [23,24,25]. In particular, magnetic chitosan microspheres have been shown to be very effective in the adsorption of methyl blue, crystal violet, Congo red, and other positive dyes [26,27,28,29]. Therefore, preparing magnetic resin from chitosan could ensure methyl orange adsorption. On the other hand, separating MRCM from the reaction medium can be effectively achieved using magnetic fields [30,31]. Futuristically, adsorbing toxic and harmful substances onto MRCM will solve some environmental problems while providing some economic and social benefits.

In this study, the adsorption potential of methyl orange on MRCM was described by its physicochemical properties and adsorption capacity. Firstly, the physical properties, swelling properties, magnetic stability, and regeneration of MRCM were studied in order to determine whether MRCM is suitable for adsorption of methyl orange. Next, this study investigates the batch adsorption, adsorption isotherm models, thermodynamics, and kinetics of MRCM adsorption of methyl orange, providing a theoretical basis for real-life wastewater treatment application.

## 2. Results and Discussion

### 2.1. Physical and Swelling Properties of MRCM

#### 2.1.1. Physical Properties of MRCM

Before preparing MRCM, chitosan must be fully dissolved so that the chitosan solution and the ferromagnetic fluid solution can be mixed evenly during the reaction. If chitosan powder and ferromagnetic fluid are mixed together beforehand, chitosan is not easily dissolved. Moreover, the ferromagnetic fluid is oxidized into dark red Fe_2_O_3_ particles, and the spherical product prepared has no magnetism. When the chitosan solution and the ferromagnetic fluid solution are mixed ultrasonically for 10 min, it allows the ferromagnetic fluid to be fully mixed in the chitosan solution. At the beginning of the reversed-phase suspension cross-linking method, the dispersion and emulsification steps should be maintained at room temperature because, in this step, the ferromagnetic fluid has not yet cooperated with chitosan. If this step adopts heating operation, the divalent iron ion is easily oxidized into trivalent iron ions, and the product magnetism will be reduced. During the pre-cross-linking process, under the condition of slow stirring, formaldehyde turns chitosan into a spherical liquid gel, and iron and chitosan undergo adsorption and coordination reactions. During the cross-linking process, it is always necessary to keep stirring at a low speed so that regular chitosan particles can be formed.

MRCM pictures measured by a three-dimensional microscopy system with super-depth of field are shown in Figure 1. MRCM is a black spherical particle (Figure 1a) with a smooth and raised surface (Figure 1b). This may be magnetic particles wrapped inside the chitosan microsphere. The MRCM particle size analysis results measured by laser diffraction particle size analyzer (Figure 2) showed that the average particle size of MRCM was approximately 208.138 μm. The 100% particle size of MRCM was less than 363.078 μm, and the particle sizes of 90%, 50%, and 10% were less than 286.032 μm, 197.515 μm, and 140.472 μm, respectively. During the preparation of MRCM, the mass ratio of added iron to chitosan was approximately 0.173:1. By measurement, the iron content in MRCM was 19.3655 mg/g. The average mass ratio of iron to magnetic particles was 0.019:1, indicating that approximately 10.98% of the ferromagnetic fluid added to MRCM. The physical properties of MRCM are shown in Table 1. MRCM contained iron ions that can bind to water molecules, so MRCM has high water content. Furthermore, during the preparation of MRCM, two reagents, formaldehyde and glutaraldehyde, were used, which impart physical properties of MRCM such as cross-linking degree and suspension aldehyde group. In addition to the coordination effect with iron ions, the free amino group of chitosan in MRCM had some residues, so MRCM had a weak alkali exchange capacity. From the above analysis, it can be seen that MRCM is a type of black spherical, porous, water-absorbing, and weak alkali exchange resin, and it can be inferred that MRCM has the ability to adsorb methyl orange.

#### 2.1.2. Results of Swelling Properties of MRCM in Different pH Solutions

A certain quantity of dry MRCM swelled for 96 h in buffer solution with different pH values. The equilibrium swelling rates are shown in Table 2. When the swelling experiment was over, MRCM in HCl-KCl buffer at pH 1.0 lost its magnetic iron particles and became larger yellow chitosan particles. The color of MRCM in other pH buffers did not change, so it was possible that MRCM was unstable in pH 1.0 solution. In addition to the pH 1.0 buffer, MRCM can also swell in other pH buffer solutions. The equilibrium swelling rates of MRCM in both Tris-HCl buffer at pH 9.0 and borax–sodium hydroxide buffer at pH 11.0 were significantly lower than those in other pH solutions (*p* < 0.05). In addition, its equilibrium swelling rate in pH 3.0 Na_2_HPO_4_—citric acid buffer—was significantly higher than that in other pH solutions (*p* < 0.05). It is indicated that MRCM can be used in solutions with pH > 1.0.

#### 2.1.3. Stability of Magnetic Particles in MRCM

The stability of Fe_3_O_4_ particles in MRCM can be determined by the determination of total iron concentration in the supernatant of MRCM soaked in different pH solutions for a certain time. The experimental results are shown in Table 3. In pH 1.0 solution, the Fe_3_O_4_ of MRCM was unstable and the Fe_3_O_4_ was completely released from the MRCM. The particles changed from black to yellow and became larger. This result was the same as that of the equilibrium swelling rate in pH 1.0 solution at 2.1.2. The release of iron ion reached the maximum at 36 h and finally stabilized at 8.904 mg/L, which indicated that MRCM could not be used in pH 1.0 solution. In pH 2.0 solution, very few iron ions were released, and the concentration of total iron ions in the supernatant was stable at 0.023 mg/L, and no color change was found. In pH 3.0–8.0 solutions, very few iron ions were released and MRCM remained unchanged. Therefore, MRCM is suitable for applications in solutions above pH 2.0.

### 2.2. Results of Batch Adsorption Studies

#### 2.2.1. Effect of pH on MRCM Adsorption of Methyl Orange

The effect of pH on the adsorption of methyl orange on MRCM was examined within 3–11 pH range (Figure 3a). Other experiments were conducted at pH 5, where the maximum adsorption capacity was obtained (*p* < 0.05). We observed that the adsorption capacity at pH 11 was significantly lower than that at other pH values (*p* < 0.05).

Methyl orange usually exists as -SO_3_^−1^ in the aqueous phase. However, at pH 11, numerous OH^-^ would compete with methyl orange anion for MRCM adsorption sites. Because of a few unfavorable factors limiting methyl orange anion more than OH^-^ (such as larger ionic volume, higher bonding energy with MRCM, and lower bonding force), the minimum adsorption capacity was obtained at pH 11. It can be seen from Figure 3a that MRCM has a good *adsorption* capacity in the pH 3–9 range, which is consistent with the conclusion obtained by Zhang et al. [32].

#### 2.2.2. Effect of Contact Time on MRCM Adsorption of Methyl Orange

Studying the adsorption at various contact times helps to determine the rate and capacity of adsorption by the adsorbent. Figure 3b shows the effect of contact time on the adsorption of methyl orange on MRCM. The adsorption equilibrium time was 140 min. It can be seen from Figure 3b that the adsorption capacities at 140, 160, and 180 min were significantly larger than those at other times (*p* < 0.05). In addition, 81.26% of the adsorption capacity was reached in the first 60 min. After this point, the adsorption rate reduced, becoming more meager with time, before reaching equilibrium after 3 h of contact.

#### 2.2.3. Effect of Temperature on MRCM Adsorption of Methyl Orange

Figure 3c illustrates changes in the adsorption capacity of MRCM against methyl orange with temperature. The adsorption capacities at 323 K and 333 K were significantly higher than those at the other three lower temperatures (*p* < 0.05). It can be concluded that the equilibrium adsorption capacity was enhanced with an increase in temperature, indicating an endothermic reaction.

#### 2.2.4. Effect of Initial Concentration on MRCM Adsorption of Methyl Orange

The effect of initial methyl orange concentration on its adsorption on MRCM was investigated by evaluating the reaction between various adsorbate concentrations and a predetermined amount of the adsorbent. The contact time and temperature were fixed at 3 h and 30 °C, respectively. The results are depicted in Figure 3d. The adsorption capacity also increased with the initial concentration of methyl orange. Because the number of effective adsorption sites was limited, competitive adsorption occurred as methyl orange initial concentration increased, favoring methyl orange adsorption. Hence, the maximum adsorption was reached at 450 mg/L initial concentration (*p* < 0.05).

### 2.3. Adsorption Isotherm Modeling

The adsorption isotherm was observed when the two-phase system containing MRCM and methyl orange solution reached equilibrium (Figure 4). Generally, in the solid–liquid system, adsorption occurs when the solute is attracted from the solution onto the solid surface. This process continues until a dynamic equilibrium is reached between the amount of the solute in the solution and that adsorbed onto the solid surface. Various isotherm and adsorption models could describe the limited distribution phenomenon of solute in the liquid–solid phase. This involves fitting the experimental data to determine the model parameters.

#### 2.3.1. Langmuir Adsorption Isotherm Model

The Langmuir adsorption isotherm for the methyl orange-MRCM system was investigated. The Langmuir constants were calculated at 293, 303, 313, 323, and 333 K. The model assumes that the adsorbate molecules are absorbed onto energy equivalent adsorption sites, and each site can accommodate a unique adsorbate molecule [33,34,35]. Therefore, the Langmuir adsorption isotherm is suitable for monolayer adsorption. The usual Langmuir equation is
(1)$$\frac{C_{e}}{Q}=\frac{1}{K_{b}\times Q_{s}}+\frac{C_{e}}{Q_{s}}$$
where *C_e_* (mg/L) is the adsorption equilibrium concentration, *Q* (mg/g) is equilibrium adsorption capacity, *Q_s_* (mg/g) is the monolayer saturated adsorption capacity of the resins, and *K_b_* is the Langmuir constant. The Langmuir adsorption isotherm parameters are given in Table 4, according to the linear plots of *C_e_*/*Q* vs. *C_e_* at various temperatures.

Table 4 shows high correlation coefficients of each linear regression >0.98, proving that the adsorption followed the Langmuir model. As the experimental temperature increased, *Q_s_* increased, indicating that higher temperature favors methyl orange adsorption onto MRCM, i.e., the adsorption was endothermic. The Langmuir isotherm adsorption characteristic could be denoted by a dimensionless equilibrium parameter (the characteristics separation coefficient *R_L_*). The *R_L_* is defined as follows:(2)$$R_{L}=\frac{1}{1+K_{b}\times C_{0}}$$
where *K_b_* is the Langmuir constant, and *C*_0_ (mg/L) is the initial concentration of methyl orange at a definite temperature [36]. Generally, the adsorption is non-favorable when *R_L_* > 1, linear when *R_L_* = 1, favorable when 0 < *R_L_* < 1, and irreversible when *R_L_* = 0. For the experimental concentration range, the *R_L_* values were 0.246, 0.193, 0.203, 0.108, and 0.067, respectively. This indicates that the adsorption of methyl orange on MRCM was favorable because 0 < *R_L_*< 1.

#### 2.3.2. Freundlich Adsorption Isotherm Model

This model is suitable for modified uniform surface adsorption [37,38,39]. It is expressed as follows:(3)$$\ln Q=\ln K_{f}+\frac{\ln C_{e}}{n}$$
where *C_e_* (mg/L) is the adsorption equilibrium concentration, *Q* (mg/g) is equilibrium adsorption capacity, *K_f_* is the relative value of adsorption capacity, and *n* is the adsorption strength. The Freundlich adsorption isotherm parameters are listed in Table 5, according to the linear plots of ln*Q* vs. ln*C_e_* at various temperatures.

Table 5 shows that the correlation coefficients of each linear regression equation were >0.85, proving that the adsorption also followed the Freundlich model. Also noticed was that the higher the temperature, the higher the *K_f_*, suggesting that temperature increase favors methyl orange adsorption onto MRCM. Overall, the values of the constant, *n*, indicate favorable adsorption, i.e., *n* > 1.

#### 2.3.3. Dubinin–Radushkevich (D-R) Adsorption Isotherm Model

The D-R isotherm adsorption model could be applied more widely than Langmuir, BET, and Redke–Prausnitz models because it is not based on the hypothesis of homogeneous adsorption sites and the equivalent adsorption potential. The D-R model is expressed as follows:(4)$$\ln Q_{e}=\ln Q_{m}-K\times E^{2}$$
where *Q_e_* (mg/g) is the equilibrium adsorption capacity, *Q_m_* (mg/g) is the theoretical saturated adsorption capacity, and *K* (mol/kJ)^2^ is the D-R adsorption constant related to the mean free adsorption energy, which indicates the energy
(5)$$E^{\prime }(\mathrm{kJ}/mol)={\left(2K\right)}^{-\frac{1}{2}}$$
required to transfer from the boundless space in solution to the surface of a solid sorbent, and *E* (kJ/mol) is the Polanyi adsorption potential [40,41]. The *E* can be calculated using the formula:(6)$$E=-R\times T\times \ln\frac{C_{e}}{C_{0}}$$
where *R* is the gas constant (8.314 J/K·mol), *T* is the absolute temperature (*K*), and *C_e_* (mg/L) is the equilibrium concentration. The D-R adsorption isotherm parameters are given in Table 6, according to the linear plots of ln*Q_e_* vs. *E*^2^ at various temperatures.

The adsorption of methyl orange on MRCM needed *E*′ values of 4.419, 5.955, 6.155, 6.967, and 7.809 kJ/mol for the 293–333 K temperature range, respectively. Generally, the bond energy number range of monolayer adsorption, hydration water adsorption, and multilayer adsorption was 4–21, 8–42, and 290–420 kJ/mol, respectively. In terms of the *E*′ value obtained from the experiment, monolayer adsorption was the dominant adsorption mode for methyl orange adsorption on MRCM. This result was consistent with the Langmuir linear fitting, suitable for describing the adsorption process.

#### 2.3.4. Change in Adsorption Potential

According to Polanyi theory, the adsorption potential (*E*) of an adsorbent molecule onto the solid surface is the power required to move the molecule from its current position in adsorption space to the boundless space. Because the concentration of the bulk phase is uniform, it shows that the effect of the adsorption force field in the bulk phase can be neglected compared with the thermal motion of the molecules in the bulk phase. The adsorption force is zero, then, the interface between bulk phase and adsorption phase is zero. Therefore, the adsorption potential of adsorbent molecule in the adsorption force field of solid surface is the power required to move the molecule from the position of isosurface to zero point in the adsorption phase. The solid–liquid adsorption potential can be expressed as Formula (6). The adsorption potential of methyl orange onto MRCM is listed in Table 7.

At a fixed temperature, the adsorption potential of methyl orange onto MRCM gradually decreased with the increase in methyl orange concentration. In addition, the adsorption potential increased with experimental temperature at a fixed initial methyl orange concentration because the adsorption was an endothermic reaction, i.e., MRCM showed high adsorption potential for methyl orange at elevated temperatures. When the temperature was constant, the adsorption isotherm showed that the adsorption improved with increased methyl orange initial concentration. However, the adsorption potential decreased with an increase in the adsorption quantity. Therefore, the higher the initial concentration of the adsorbate, the better the adsorption, albeit the lower the adsorption potential. Two reasons could be attributed to this decreased adsorption potential: (1) the adsorption initially occurred on the sites with the highest attraction force on the heterogeneous MRCM surface; and (2) the surface attraction force decreased with the rise of surface coverage when the adsorbed quantity reached a certain value.

### 2.4. Adsorption Thermodynamics of Methyl Orange onto MRCM

#### 2.4.1. Change in Adsorption Enthalpy

The Clausius–Clapeyron equation, used for the thermodynamic investigation, is as follows:(7)$$\ln C_{e}=\frac{\Delta H}{R\times T}+K$$
where *R* is the gas constant (8.314 J/K·mol), *T* is the absolute temperature (*K*), *C_e_* (mg/L) is the adsorption equilibrium concentration, Δ*H* is the isosteric adsorption enthalpy (kJ/mol), and *K* is the adsorption equilibrium constant [42]. The adsorption isosteres ln *C_e_* − 1/*T* (ln *C_e_* given by Langmuir equation) could be obtained by using the adsorption isotherm of methyl orange onto MRCM at different temperatures. Figure 5 illustrates the adsorption enthalpy of adsorption methyl orange onto MRCM. The correlation was excellent at all the initial concentrations. Therefore, in the temperature range of this study, it is reasonable that the hypothesis Δ*H* in the derivation of this formula is independent of *T*. The slope (Δ*H*/*R*) corresponding to adsorption quantity and the isosteric enthalpy Δ*H* could be calculated by the linear regression method (Table 8). As *Q* increased from 40 to 70 mg/g, Δ*H* increased. The methyl orange adsorption onto MRCM was confirmed to be endothermic. That is, an increase in temperature favors the adsorption, denoted by the positive Δ*H* values.

The enthalpy change of the adsorption process is caused mainly by the adsorption heat of methyl orange, the desorption heat of solvent water, the dilution heat of solution caused by the adsorption of methyl orange, and the resolvation heat of solute molecules. The positive and negative enthalpy change depends on the contrast between the total exothermic effect and the total endothermic effect. In this experimental system, the total exothermic effect came mainly from the adsorption heat of methyl orange, and the total endothermic effect came mainly from the activation energy of chitosan molecular chain. The adsorption of methyl orange onto MRCM may have led to the crystallinity decrease of MRCM and the destruction of intramolecular and intermolecular hydrogen bonds. The molecular conformation of chitosan changed to the unstable state. All of these processes consumed a lot of energy. When the temperature increased, the amount of methyl orange adsorbed by MRCM increased, and more energy was consumed.

On the other hand, methyl orange desorption from the MRCM was an endothermic process because the heat of adsorption was less than the total endothermic effect.

#### 2.4.2. Change in Adsorption Free Energy

While the adsorption isotherm data of MRCM fitted the Freundlich equation, the adsorption free energy did not correlate with *Q* in the low adsorbate concentration. The adsorption free energy value can be obtained from the equation [43]
(8)ΔG=−n×R×T

The adsorption free energy change was the embodiment adsorption driving force; a negative value indicates that spontaneous adsorption would proceed. The adsorption free energy change in the current study had low temperature dependence (Table 8). This result shows the physical characteristics of the adsorption and the entropy compensation in the low adsorption capacity. Usually, the adsorption free energy change of physical adsorption (from −20 to 0 kJ/mol) is less than that of chemical adsorption (from −400 to −80 kJ/mol). The Δ*G* data in Table 8 showed that methyl orange collection on MRCM was a spontaneous, physical adsorption.

#### 2.4.3. Change in Adsorption Entropy

The adsorption entropy change can be calculated according to the Gibbs–Helmoltz equation [44]:(9)$$\Delta S=\frac{\left(\Delta H-\Delta G\right)}{T}$$

The adsorption free energy change was negative (Δ*G* < 0) because the adsorption was spontaneous, i.e., Δ*H* − *T*Δ*S* < 0, Δ*S* > 0, and Δ*S* > |Δ*H*/*T*| were due to Δ*H* > 0. Therefore, the adsorption was an entropy-increasing process promoted by enthalpy. In the experimental range, the adsorption entropy change was positive, meaning that adsorption predominates the transfer of methyl orange onto the MRCM surface. At the same time, negative Δ*S* values indicate a restricted transfer of methyl orange onto the resin surface.

Furthermore, when methyl orange adheres to MRCM, water molecules might be desorbed from the MRCM, eluting into the solution. This suggestion was displayed as the increase in desorption entropy change (positive value). Because the displaced water molecules were more than the adsorbed methyl orange, the total entropy change was positive, indicative of a larger molar volume of water than that of methyl orange.

### 2.5. The Adsorption Kinetics Behavior of Methyl Orange onto MRCM

We investigated four main aspects of methyl orange adsorption kinetics onto MRCM [45]. The first was bulk diffusion, in which methyl orange is transferred from the bulk phase solution to the film edge of the adsorbent surface. The second was film diffusion, in which methyl orange is transferred from the film edge to the adsorbent. The third was intra-particle diffusion, in which methyl orange is transferred from the adsorbent surface to the active site of inner particle pores. The last was a chemical reaction, in which methyl orange is adsorbed onto the adsorbent via ion-exchange.

To evaluate the kinetic mechanism of methyl orange adsorption onto MRCM, the adsorption kinetics were investigated by the pseudo-first-order Lagergren model, the pseudo-second-order kinetics equation, the liquid film diffusion model, and the intra-particle diffusion model.

#### 2.5.1. The Pseudo-First-Order Lagergren Model

The pseudo-first-order Lagergren model is expressed as follows:(10)$$\ln(Q_{e}-Q_{t})=\ln Q_{e}-\frac{K_{1}\times t}{2.303}$$
where *Q_e_* and *Q_t_* are the adsorption quantities (mg/g) at the equilibrium time and *t* time, respectively, and *K*_1_ (1/min) is the rate constant of the pseudo-first-order kinetics [46]. The Lagergren model parameters are listed in Table 9, according to the linear plots of ln(*Q_e_* − *Q_t_*) vs. *t* at various contact times. Generally, the pseudo-first-order Lagergren model did not fit well with the adsorption data. In this study, the model could describe only 120 min of the adsorption. In addition, the equilibrium adsorption quantity of the theoretical calculation was much less than that of the experimental data. Therefore, the pseudo-first-order Lagergren model could not be used to illustrate methyl orange adsorption onto MRCM.

#### 2.5.2. The Pseudo-Second-Order Kinetics Equation

The equation
(11)$$\frac{t}{Q_{t}}=\frac{1}{K_{2}\times Q_{e}{}^{2}}+\frac{t}{Q_{e}}$$
calculates the rate constant of the pseudo-second-order kinetics adsorption interaction [47], where *Q_e_* and *Q_t_* are the adsorption quantity (mg/g) at equilibrium time and *t* time, respectively, while *K*_2_ (g/mg·min) is the rate constant. The pseudo-second-order kinetics equation parameters are given in Table 9, according to the linear plots of *t*/*Q_t_* vs. *t* under various times. The pseudo-second-order kinetics better describes methyl orange adsorption onto MRCM than the pseudo-first-order model as denoted by the better regression correlation coefficient and the lower rate constant of the former.

#### 2.5.3. Liquid Film Diffusion Model

It is uncertain whether the liquid film diffusion is a rate-limiting step in the adsorption. The expression of the liquid film diffusion model is
(12)$$\ln(1-F)=-K_{fd}\times t+C$$
where *F* (*F* = *Q_t_*/*Q_e_*) is the integration of equilibrium adsorption quantity, and *K_fd_* (1/min) is the rate constant [48]. The model parameters are listed in Table 9, according to the linear plots of −ln(1 − *F*) vs. *t*. Generally, when the adsorbate speed outside the adsorbent is less than that inside the adsorbent pores, the liquid film diffusion would be the rate-limiting step of the adsorption. Here, the liquid film diffusion was the rate-limiting step because the correlation coefficient was >0.91, and the rate constant was less than that of the intra-particle diffusion model (discussed next).

#### 2.5.4. The Intra-Particle Diffusion Model

The intra-particle diffusion model also describes whether the intra-particle diffusion is the rate-limiting step of adsorption. The model expression is as given:(13)$$Q_{t}=K_{id}\times t^{0.5}+C$$
where *Q_t_* is the adsorbed quantity (mg/g) at time *t*, and *K_id_* (mg/g × *t*^0.5^) is the rate constant [49]. The intra-particle diffusion model parameters are provided in Table 9, according to the linear plots of *Q_t_* vs. *t*^0.5^. Usually, when the adsorbate transfer rate outside the adsorbent is higher than that inside the adsorbent pores, the intra-particle diffusion is regarded as the rate-limiting step in the whole adsorption. Here, the rate constant of the intra-particle diffusion model was higher than that of the liquid film diffusion model. Therefore, intra-particle diffusion would be completed faster than liquid film diffusion. Therefore, intra-particle diffusion was not the rate-limiting step in the adsorption.

### 2.6. Results of Regeneration of Methyl Orange Adsorption onto MRCM

After methyl orange adsorption onto MRCM, MRCM was vacuum-filtered. Then, MRCM was washed successively with 2.0 mol/L NaCl solution, distilled water, 0.05 mol/L NaOH solution, and distilled water, and the process of adsorption and washing was repeated five times.

When MRCM was washed by oscillating with 2 mol/L NaCl solution, the color of the NaCl solution changed to orange, indicating that methyl orange was desorbed from MRCM. Next, when MRCM was washed with 0.05 mol/L NaOH solution, the NaOH solution gradually turned yellow. This indicated that methyl orange could not be completely desorbed from MRCM by NaCl solution, and it needed stronger OH^−1^ ion to desorb methyl orange. This is also the reason why the adsorption capacity of MRCM to methyl orange was smaller when the pH value was 11 (see Section 2.1.2 Results). The MRCM adsorbed methyl orange was washed with NaCl solution, distilled water, NaOH solution, and distilled water, in turn, and then the adsorption and desorption experiments were carried out 5 times. The adsorption capacities of the five repeated experiments were 20.064 mg/g, 19.751 mg/g, 19.352 mg/g, 18.935 mg/g, and 18.918 mg/g, respectively. Then, the adsorption capacity of MRCM remained 94.29% after 5 cycles of adsorption—regeneration. Magnetic chitosan microspheres can also be recycled in the two studies of adsorption of phenolic substances by magnetic chitosan hydroxyapatite microspheres and adsorption of heavy metals by magnetic microspheres—MnFe_2_O_4_/chitosan [50,51]. It can be concluded that MRCM is a type of environment-friendly adsorption material which can be recycled and reused.

## 3. Materials and Methods

### 3.1. Materials

Chitosan, with 95% degree of deacetylation (%DD), was provided by Lizhong Chitosan Company Limited, Qingdao, China, and it was used without any purification. Ferric sulfate, ferrous sulfate, ammonia, and methyl orange (Sodium p-dimethyl amino azobenzene sulphonate) were analytical reagents produced by Sinopharm Chemical Reagent Co., Ltd., Shanghai, China. All other reagents were of analytical grade and used as received.

### 3.2. Preparation of MRCM

The MRCM was prepared by reversed-phase suspension cross-linking polymerization. The water-based ferromagnetic fluid was added to the 10% (*w*/*v*) chitosan acetic acid (2% *v*/*v*) solution in a 25% (*v*/*v*) proportion. The mixture was stirred with a glass rod and blended ultrasonically for 10 min to a black solution. Then, the mixture was added to liquid paraffin (1:1, *v*/*v*) and stirred for 10 min. An emulsion was formed when adding span-80 (1.2%, *w*/*v*) and acetic ester (10%, *v*/*v*) under stirring with the speed of 300 r/min for 20 min at room temperature. At 40 °C, the cross-linking agent, formaldehyde (8%, *v*/*v*) was added to the emulsion and spun at 300 r/min for 30 min. When the temperature rose to 50 °C, the cross-linking agent glutaraldehyde (4% *v*/*v*) was added, and the pH was adjusted to 7.5 with 2 mol/L sodium hydroxide, followed by 175 r/min stirring for 3 h. The MRCM was then collected and rinsed with petroleum ether, acetone, ethanol, and distilled water. The particles were then dried in a DZX-6 vacuum drying oven (50 °C) for approximately 48 h before storing them at room temperature.

### 3.3. Determination of Physical Properties of MRCM

#### 3.3.1. Determination of Cross-Linking Degree

The cross-linking degree was determined according to the method of You et al. [52]. The amount of 0.25 g (*W*_1_) of MRCM was soaked in 2% (*v*/*v*) acetic acid solution for 24 h. Then, acetic acid solution was removed from MRCM using vacuum infiltration. MRCM was dried at 105 °C for 4 h and weighed (*W*_2_). The cross-linking degree can be calculated by the following formula:(14)$$\xi (\%)=\frac{W_{1}-W_{2}}{W_{1}}\times 100\%$$

#### 3.3.2. Determination of Moisture Content

The moisture content was determined according to the method of Sumthane et al. [53]. The amount of 0.1 g of MRCM was swelled fully with distilled water. Then, water was removed from MRCM using vacuum infiltration. The surface moisture of MRCM was extracted and weighed to obtain *G*_1_. MRCM was dried at 105 °C for 4 h and weighed to obtain *G*_2_. The moisture content can be calculated by the following formula:(15)$$H(\%)=\frac{G_{1}-G_{2}}{G_{1}}\times 100$$

#### 3.3.3. Determination of Packing Density

The packing density was determined according to the method of Yu et al. [29]. The packing density is the weight of MRCM per unit volume, including the volume of the ball’s skeleton, the volume of the holes, and the volume of the gaps between the balls. The amount of 2.0 mL (*V_P_*) of MRCM was placed in a 10 mL measuring cylinder and weighed to obtain *W*. The packing density can be calculated by the following formula:(16)$$\rho_{P}(g/\mathrm{mL})=\frac{W}{V_{P}}$$

#### 3.3.4. Determination of Skeleton Density

The skeleton density was determined according to the method of Yu et al. [54]. The amount of 5.0 mL of N-heptane was placed in a 10 mL measuring cylinder and weighed to obtain *W*_1_. Then, N-heptane was poured out. The amounts of 0.1 g (*W*) of MRCM and 2.0 mL of N-heptane were placed in this measuring cylinder for 2 h. Then, N-heptane was added to the 5.0 mL scale of the measuring cylinder and weighed to obtain *W*_2_. The skeleton volume can be calculated by the following formula:(17)$$V_{T}=\frac{W_{1}-W_{2}+W}{dt_{}}$$
where *dt* is N-heptane density, and its value is 0.6830 g/cm^3^. Then, skeleton density can be calculated by the following formula:(18)$$\rho_{T}({g/\mathrm{cm}}^{3})=\frac{W}{V_{T}}$$

#### 3.3.5. Determination of Porosity Value

The porosity value was determined according to the method of Yu et al. [54]. Porosity value can be calculated by the following formula:(19)$$P=\frac{\rho_{T}\times H}{\rho_{T}\times H+1-H}$$

#### 3.3.6. Determination of Suspended Aldehyde Group Content

The suspended aldehyde group content was determined according to the method of Ren et al. [55]. The amount of 0.1 g of MRCM was swelled fully with distilled water. Then, water was removed from MRCM using vacuum infiltration. The amount of 10.0 mL hydroxylamine reagent was added to MRCM and oscillated at room temperature for 1 h. Then, two drops of 0.05% bromophenol blue indicator were added and titrated to the end point with standard hydrochloric acid solution (0.02 mol/L). Suspended aldehyde group content can be calculated by the following formula:(20)$$\mathrm{Suspended}\;\mathrm{aldehyde}\;\mathrm{group}\;\mathrm{content}\;(\mathrm{mmol}/g)=\frac{N\times (V_{0}-V_{1})}{W}$$
where *N* (mol/L) is the standard concentration of hydrochloric acid solution, *V*_0_ (mL) is the blank consumption of standard hydrochloric acid volume, *V*_1_ (mL) is the MRCM consumption standard hydrochloric acid volume, and *W* (g) is MRCM quantity.

#### 3.3.7. Determination of Exchange Capacity of Weak Base

The exchange capacity of weak base was determined according to the method of Ren et al. [55]. The amount of 0.1 g of MRCM was swelled fully with distilled water. Then, water was removed from MRCM using vacuum infiltration. The amount of 20.0 mL standard hydrochloric acid solution (0.05 mol/L) was added to MRCM and oscillated at room temperature for 1 h. Then, two drops of 0.2% phenolphthalein indicator were added to 15.0 mL of supernatant and titrated to the end point with standard sodium hydroxide solution (0.05 mol/L). Exchange capacity of weak base can be calculated by the following formula:(21)$$\mathrm{Exchange}\;\mathrm{capacity}\;\mathrm{of}\;\mathrm{weak}\;\mathrm{base}\;(\mathrm{mmol}/g)=\frac{N_{1}\times V_{1}-N_{2}\times V_{2}\times \frac{20}{15}}{W}$$
where *N*_1_ (mol/L) is the standard concentration of hydrochloric acid solution, *V*_1_ (mL) is standard volume of hydrochloric acid solution, *N*_2_ (mol/L) is the standard sodium hydroxide solution concentration, *V*_2_ (mL) is standard volume of sodium hydroxide solution, and *W* (g) is MRCM quantity.

### 3.4. Measurement of Adsorption Capacity

First, 0.1 g of MRCM was swelled with distilled water before placing it in a 100 mL conical flask. Water was removed from MRCM using vacuum infiltration. Then, 40 mL methyl orange solution was added to the flask. The mixture was oscillated for 3 h at a specific temperature. In the end, the absorbance of the reaction solution was measured using the UNICO 2102 UV-Vis spectrophotometer at 464 nm. The adsorption capacity can be calculated by the following formula:(22)$$Q=\frac{(C_{0}-C)\times V}{W}$$
where *Q* (mg/g) is the adsorption capacity, *C*_0_ (mg/L) is the methyl orange concentration before adsorption, *C* (mg/L) is the methyl orange concentration after adsorption, *V* (L) is the reaction solution volume, and *W* (g) is the quantity of MRCM.

### 3.5. Batch Adsorption Studies

The adsorption of 40 mL methyl orange solutions of 50–500 mg/L concentrations using 0.1 g MRCM was carried out at 293, 303, 313, 323, and 333 K for a certain time and pH (3–11). Upon the completion of the reaction, the absorbance of the reaction solution was measured at 464 nm, and the equilibrium adsorption capacity was calculated. Then, the influencing factors and thermodynamic properties of the adsorption were evaluated.

### 3.6. Kinetic Studies

First, 40 mL of methyl orange solution (50 mg/L) was added to 0.1 g MRCM in a pre-cleaned 50 mL conical flask. The mixture was oscillated in a water bath (75 r/min) at 303 K. The absorbance of the solution was determined, and the adsorption capacity was estimated after a predetermined contact time.

### 3.7. Statistical Analysis

All experiments were performed in triplicate and data were expressed as means and standard deviations. The SPSS Statistics 17.0 software was used to analyze the variance of the results with the method of least significant difference (LSD).

## 4. Conclusions

On the basis of the study of the physical properties, swelling properties, magnetic stability, and regeneration of MRCM, the isothermal adsorption model, thermodynamics, and kinetics of methyl orange onto MRCM were investigated in this paper. According to the physicochemical properties of MRCM, MRCM is a type of porous, black spherical, water-absorbent, and swelling resin and has exchange capacity of weak base. When applied to solutions with pH greater than 2.0, the magnetic particles in MRCM are stable. Moreover, MRCM has the ability to be reused at least five times. The batch adsorption studies showed that the adsorption capacities increase with the contact time, temperature, and the initial concentration of methyl orange. However, it changes slightly in the solution with different pH, from 3 to 9. The Langmuir isotherm adsorption model better describes the adsorption mechanism MRCM adsorption of methyl orange. The results of adsorption thermodynamics showed that the adsorption was thermodynamically endothermic and spontaneous. In addition, the results of adsorption kinetics revealed that the adsorption was controlled by liquid–film diffusion dynamics, following the pseudo-second-order kinetic model. Thus, it is concluded that MRCM is suitable for the removal of methyl orange dyes from water in terms of its physical and chemical properties and adsorption thermodynamics. In addition, it is hopeful to develop MRCM as a type of sorbent for removing cationic dyes similar to methyl orange from water because MRCM is an environment-friendly sorbent and can be regenerated and reused. In a word, this study is beneficial for promoting further research for toxic dyestuffs pollution control. In future research, magnetic chitosan microspheres with different functional groups grafted on MRCM will be developed, and it is expected to extend its application to adsorption of heavy metals, anion/cation dyes, active substances, immobilized enzymes, and other fields.

## Figures and Tables

**Figure 1 ijms-23-13839-f001:**
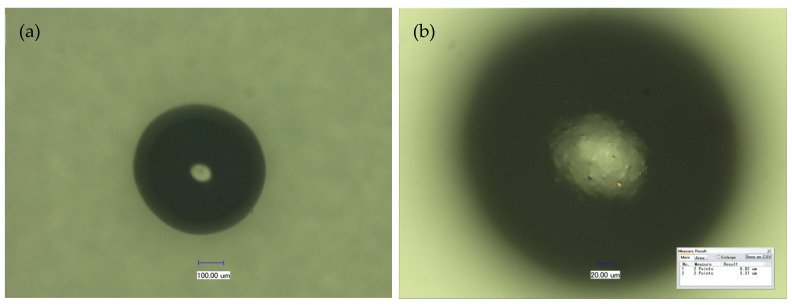
MRCM picture (**a**) and MRCM surface photograph (**b**).

**Figure 2 ijms-23-13839-f002:**
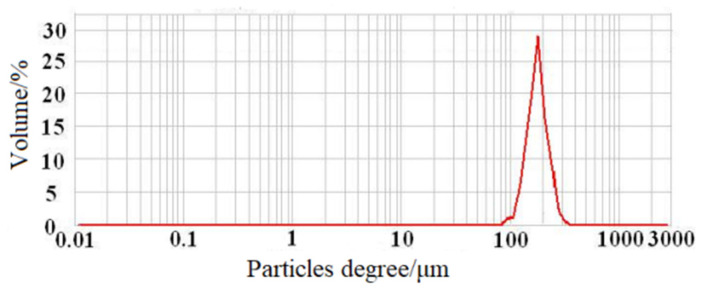
Normal distribution of MRCM particle size.

**Figure 3 ijms-23-13839-f003:**
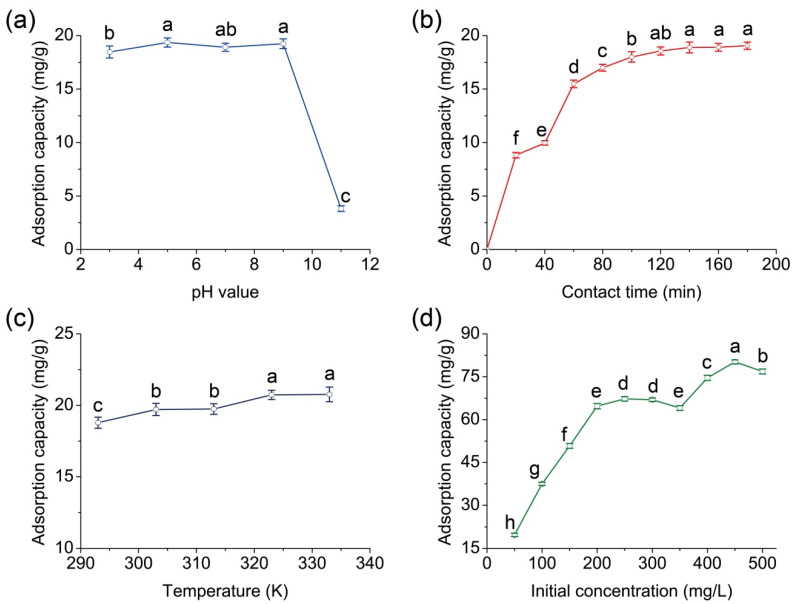
Effects of pH (**a**), contact time (**b**), temperature (**c**), and initial concentration (**d**) on MRCM adsorption of methyl orange at 303 K. (The reaction conditions in (**a**–**c**) are: the concentration of methyl orange is 50 mg/L, and the amount of MRCM is 0.1 g. The reaction conditions in (**d**) are: the reaction temperature is 30 °C, and the amount of MRCM is 0.1 g. Different letters in each of Figure (**a**–**d**) mean significant difference at 0.05 level (*p* < 0.05, *n* = 3).

**Figure 4 ijms-23-13839-f004:**
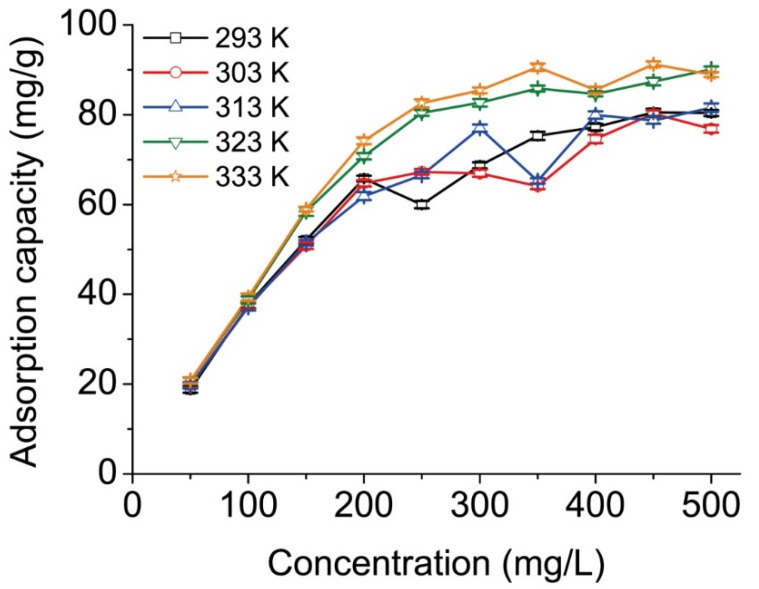
Adsorption isotherms of methyl orange onto MRCM.

**Figure 5 ijms-23-13839-f005:**
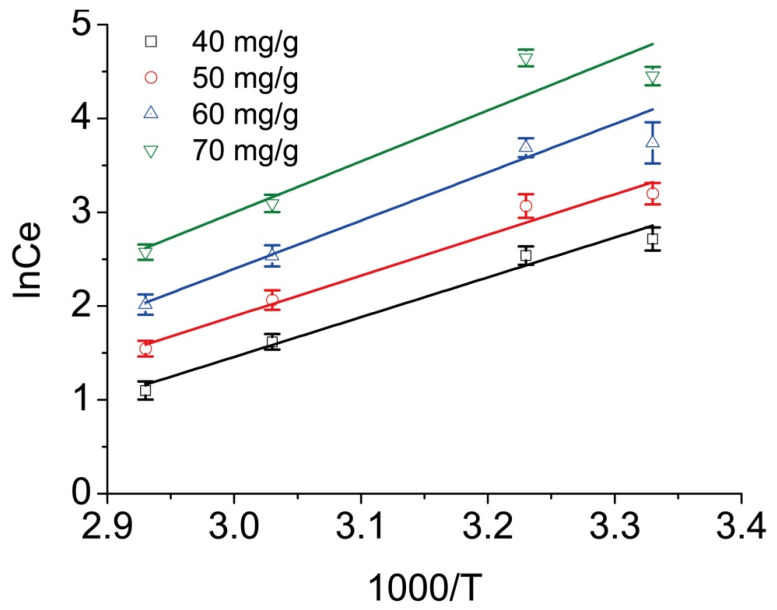
Determination of enthalpy of methyl orange adsorption onto the MRCM (The different colored lines represent the adsorption isograms when the isosorption amounts are 40, 50, 60 and 70 mg/g, respectively).

**Table 1 ijms-23-13839-t001:** Physical properties of MRCM.

Moisture Content (%)	Cross-Linking Degree (%)	Pile-Up Density (g/mL)	Pore Degree	Suspension Aldehyde Group (mmol/g)	Weak Alkali Exchange Capacity (mmol/g)
52.106 ± 2.084	10.400 ± 0.400	0.922 ± 0.058	0.527 ± 0.045	0.267 ± 0.012	1.393 ± 0.095

**Table 2 ijms-23-13839-t002:** Swelling properties of MRCM in different pH solutions.

	pH 1.0	pH 3.0	pH 5.0	pH 7.0	pH 9.0	pH 11.0	pH 13.0
Swelling rates	0.216 ^a,b^ ± 0.008	0.228 ^a^ ± 0.020	0.205 ^a,b^ ± 0.008	0.193 ^b,c^ ± 0.018	0.165 ^c^ ± 0.008	0.167 ^c^ ± 0.011	0.208 ^a,b^ ± 0.034

Note: Different superscript letters of different pH values mean significant differences at 0.05 level (*p* < 0.05, *n* = 3).

**Table 3 ijms-23-13839-t003:** Total iron concentration released from MRCM in solutions with different pH values.

Time (h)	pH 1.0	pH 2.0	pH 3.0	pH 4.0	pH 5.0	pH 6.0	pH 7.0	pH 8.0
12	6.327 ± 0.106	0.083 ± 0.008	0.000	0.000	0.000	0.000	0.000	0.000
24	6.257 ± 0.112	0.000	0.000	0.000	0.000	0.000	0.000	0.000
36	9.414 ± 0.098	0.003 ± 0.001	0.000	0.000	0.000	0.000	0.000	0.000
48	6.726 ± 0.101	0.043 ± 0.005	0.003 ± 0.001	0.003 ± 0.001	0.000	0.013 ± 0.001	0.000	0.000
60	8.854 ± 0.124	0.023 ± 0.002	0.000	0.000	0.000	0.000	0.000	0.000
72	8.564 ± 0.116	0.023 ± 0.001	0.000	0.000	0.000	0.000	0.003 ± 0.001	0.013 ± 0.002
84	8.934 ± 0.131	0.023 ± 0.001	0.000	0.000	0.000	0.000	0.000	0.000
96	8.904 ± 0.123	0.023 ± 0.002	0.000	0.000	0.000	0.000	0.000	0.000

**Table 4 ijms-23-13839-t004:** Langmuir constants of methyl orange adsorption onto MRCM.

T(K)	Regression Equation	*Q_s_* (mg/g)	*K_b_* (L/mg)	R^2^
293	*C_e_*/*Q* = 0.012 *C_e_* + 0.1961	83.333	0.061	0.9911
303	*C_e_*/*Q* = 0.0128 *C_e_* + 0.1546	78.125	0.083	0.9862
313	*C_e_*/*Q* = 0.0123 *C_e_* + 0.1567	81.301	0.078	0.9845
323	*C_e_*/*Q* = 0.0111 *C_e_* + 0.0702	90.090	0.158	0.9988
333	*C_e_*/*Q* = 0.0111 *C_e_* + 0.0418	90.090	0.266	0.9988

**Table 5 ijms-23-13839-t005:** Freundlich constants for the adsorption of methyl orange onto MRCM.

T(K)	Regression Equation	*K_f_*	*n*	R^2^
293	ln*Q* = 0.2685 ln*C_e_* + 2.943	18.973	3.724	0.8587
303	ln*Q* = 0.2269 ln*C_e_* + 3.1178	22.597	4.407	0.9328
313	ln*Q* = 0.2387 ln*C_e_* + 3.0963	22.116	4.189	0.9530
323	ln*Q* = 0.2195 ln*C_e_* + 3.3598	28.783	4.556	0.9169
333	ln*Q* = 0.2068 ln*C_e_* + 3.4614	31.862	4.836	0.8871

**Table 6 ijms-23-13839-t006:** D-R constants for methyl orange adsorption onto MRCM.

T(K)	Regression Equation	*Q_m_* (mg/g)	*K* (mol/kJ) ^2^	R^2^
293	ln*Q_e_* = −0.0256 *E*^2^ + 4.4130	82.517	0.026	0.8418
303	ln*Q_e_* = −0.0141 *E*^2^ + 4.3079	74.284	0.014	0.9631
313	ln*Q_e_* = −0.0132 *E*^2^ + 4.3334	76.203	0.013	0.9490
323	ln*Q_e_* = −0.0103 *E*^2^ + 4.5217	91.992	0.010	0.9885
333	ln*Q_e_* = −0.0082 *E*^2^ + 4.5506	94.689	0.008	0.9478

**Table 7 ijms-23-13839-t007:** Adsorption potential of methyl orange onto MRCM.

T (K)	Regression Equation	*Q_m_* (mg/g)	*K* (mol/kJ) ^2^	R^2^
293	ln*Q_e_* = −0.0256 *E*^2^ + 4.4130	82.517	0.026	0.8418
303	ln*Q_e_* = −0.0141 *E*^2^ + 4.3079	74.284	0.014	0.9631
313	ln*Q_e_* = −0.0132 *E*^2^ + 4.3334	76.203	0.013	0.9490
323	ln*Q_e_* = −0.0103 *E*^2^ + 4.5217	91.992	0.010	0.9885
333	ln*Q_e_* = −0.0082 *E*^2^ + 4.5506	94.689	0.008	0.9478

**Table 8 ijms-23-13839-t008:** Thermodynamic properties of the methyl orange adsorption onto MRCM.

*Q*	Δ*H*	Δ*G* (kJ/mol)				Δ*S* (kJ/mol × K)			
(mg/g)	(kJ/mol)	293 K	303 K	323 K	333 K	293 K	303 K	323 K	333 K
40	33.485	−9.091	−11.102	−12.235	−13.389	0.145	0.147	0.142	0.141
50	34.765	−9.091	−11.102	−12.235	−13.389	0.150	0.151	0.146	0.145
60	37.090	−9.091	−11.102	−12.235	−13.389	0.158	0.159	0.153	0.152
70	42.756	−9.091	−11.102	−12.235	−13.389	0.177	0.178	0.170	0.169

**Table 9 ijms-23-13839-t009:** The adsorption kinetics model parameters for the adsorption methyl orange onto MRCM.

Kinetics Models	*K*	*Q_e_* (mg/g)	R^2^	Equation
Pseudo–first–order Lagergren model	0.046 (1/min)	16.150	0.9612	ln(*Q_e_* − *Q_t_*) = 2.7891 − 0.0199t
Pseudo-second-order kinetics equation	0.001 (g/mg·min)	23.585	0.9799	t/*Q_t_* = 1.5608 + 0.0424t
Liquid film diffusion model	0.0191 (1/min)	-	0.9492	ln(1 − *F*) = 0.0191t + 0.3042
Intra-particle diffusion model	1.229 (mg/g·t^0.5^)	-	0.8603	*Q_t_* = 1.2293t^0.5^ + 4.2819

## Data Availability

Not applicable.

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
