# Peer review of "Isotherm, Thermodynamics, and Kinetics of Methyl Orange Adsorption onto Magnetic Resin of Chitosan Microspheres"

_ijms, 2022, doi:10.3390/ijms232213839_

Round 1

Reviewer 1 Report

Abstract

This section is vague. Please present your main results.

Introduction

The topics must be better linked.

Materials and methods

A statistical analysis section is missing.

SEM of the microspheres??

Results and discussion

The results must present a suitable statistical analysis.

Please add average values plus standard deviation. Please add different superscript letters for significant differences. Moreover, please revise the discussion in accordance.

The goodness of fit of each model was only assessed by the coefficient of determination??Please clarify.

Conclusion

Please do not repeat your results and focus on your main conclusions.

Author Response

Dear reviewer,

We have received the comments of the reviewers. As a member of our group, I express my sincere appreciation here to editors and specialist reviewers for your instructions and helps to this article. We’ll revise our manuscript point by point according to the advice from the International Journal of Molecular Sciences Editorial Team and reviewers. The revision explanations are listed in “revision explanation text”. Moreover, the revised manuscript has been proofread very closely for mistakes and grammatical errors.  

I admire you for your scientific and serious attitude.

With Best Regards,

Prof. Lina Yu

Reviewer 2 Report

This manuscript focused on the dye removal application of magnetic resin of chitosan microspheres, prepared by reversed-phase suspension cross-linking polymerization. There are several works like this work by using magnetic nanoparticles with chitosan. But this work mainly focused on the adsorption isotherms.

There are some comments

1 Introduction should be expanded by citing more literature

2 The quality of the figures is low.

3 The standard deviation on the plots can not be seen or missing. Should be added.

Author Response

(The authors gave the same response as above.)

Round 2

Reviewer 2 Report

It can be accepted.

Author Response

Dear reviewer,

    As a member of our group, I express my sincere appreciation here to specialist reviewer for your instruction and help to this article. I admire you for your scientific and serious attitude.

With Best Regards,

Prof. Lina Yu